# Non-Invasive Prenatal Testing (NIPT) Implementation in Japan: A Comparison with the United Kingdom, Germany, Italy, Sweden, and Taiwan

**DOI:** 10.3390/ijerph192416404

**Published:** 2022-12-07

**Authors:** Mayo Takahashi, Le Khac Linh, Ahmad M. Sayed, Atsuko Imoto, Miho Sato, Kadek Agus Surya Dila, Nguyen Tien Huy, Kazuhiko Moji

**Affiliations:** 1School of Tropical Medicine and Global Health, Nagasaki University, Nagasaki 852-8523, Japan; 2College of Health Sciences, VinUniversity, Hanoi 100000, Vietnam; 3Department of Organic Chemistry, College of Pharmacy, Al-Azhar University, Cairo 11651, Egypt; 4Giri Emas Hospital, Singaraja City 81171, Buleleng, Bali, Indonesia; 5School of Global Humanities and Social Sciences, Nagasaki University, Nagasaki 852-8523, Japan

**Keywords:** NIPT, prenatal screening, prenatal diagnosis, perinatal care

## Abstract

Introduction: The Non-Invasive Prenatal Testing (NIPT) guideline was issued and applied in 2013 by the Japanese Medical Association. Since being issued, the NIPT practice in Japan still has some problems related to indication, access, cost coverage and uniformity. Therefore, our study aimed to identify the Japanese challenges of adopting NIPT into prenatal diagnosis by comparing the system and process with other countries. Method: The United Kingdom, Germany, Italy, Sweden, and Taiwan were purposefully selected for comparison. All the countries, including Japan, introduced NIPT. The literature and information searches were conducted using PubMed, SCOPUS, Google Scholar, CiNii and Google searching engine. Results: The process of NIPT in Japan was very different from the other countries. Japan is the only country that indicated NIPT for only pregnant women over 35 years old in certificated facilities and did not have a policy regarding providing information on prenatal screening and NIPT to all women. Japan also did not have a policy regarding abortion due to fetal abnormalities. The practice of NIPT guidelines is different between non-certified and certified facilities. NIPT fee was the highest in Japan and was not covered by insurance. Conclusion: Pregnant women in Japan suffered from disparities in information access, economic burden, geographic location, and practice of NIPT guidelines between the certified and the non-certified facilities. Pregnant women-centered prenatal diagnosis policy, including NIPT, should be established in Japan by learning cases from other countries.

## 1. Introduction

Non-Invasive Prenatal Testing (NIPT) is a challenging technology that analyzes the maternal blood for cell-free fetal DNA (cffDNA) to estimate the possibility of fetal aneuploidy [1]. It was first introduced in Hong Kong in 2011 and became available in more than 60 countries by 2014 [2]. The NIPT analysis is varied according to the supplier; for instance, it can detect trisomy 13, 18, 21, and aneuploidies of the sex chromosome [3]. Yet, NIPT is not advised as the first choice for screening; it is used clinically as a second-tier test in high-risk pregnancies [1,3,4,5]. For instance, the NIPT is performed only after a positive result of enhanced first-trimester screening or maternal serum screening. Using NIPT as a second-tier is cost-saving compared to using conventional prenatal scanning methods, but more cases of chromosomal anomalies would be detected if NIPT was implemented as a first-tier test [3,6].

The regulatory authorities in Japan approved NIPT in 2013; then, the NIPT guideline was issued by the Japanese Society of Obstetrics and Gynecology (JSOG) in the same year [7,8]. Only medical facilities certified by the Japanese Medical Association could offer adequate genetic consulting and were allowed to perform NIPT [7]. From April 2013 to March 2016, 55 nationwide certified facilities performed 30,613 tests [7]. Up to 2019, 92 accredited facilities provided NIPT in Japan. These certified facilities completed around 3500 tests per quarter in 2016, and the numbers remained steady over the years [9].

The non-certified facilities (NCFs) were allowed to perform NIPT in 2016. On 1 October 2019, Japan had 55 non-certified facilities providing NIPT, most of them located in the main cities and having served 3500 tests or more each quarter since 2016 [9]. These non-certified facilities are usually advertised on the internet as setting low prices, lacking any age restrictions, being available for gender diagnosis and whole chromosome testing, and delivering the results by mailing system. However, the doctors who indicated and consulted results for a pregnant woman could internal medicine doctors, plastic surgeons, and cosmetic surgeons [9]. Besides the lack of certification, these health care professionals (HCPs) are not specialized in genetic consulting. This raised many concerns about conducting NIPT at the non-certified facility. Forty-four certified facilities reported harmful consequences for pregnant women, such as inappropriate results of genetic counselling and lack of follow-up care such as amniocentesis and chorionic villus sampling (CVS) resulting from NIPT at the non-certified facilities [9]. In March 2019, JSOG published “Guidelines for New Prenatal Genetic Testing Using Maternal Blood” that indicated the problems of non-certified facilities. The guidelines determined such problems as (1) undertaking NIPT without sufficient awareness/understanding, (2) misunderstanding the meaning of NIPT results, and (3) using a mass screening test to detect fetal diseases [10]. The Japanese government needs to establish a rigid policy and new guidelines for using NIPT to ensure that it is used effectively and safely.

The objective of this study was to identify the Japanese challenges of adopting NIPT effectively in the prenatal diagnosis by comparing it with other countries in four domains: (1) policy on prenatal screening; (2) abortion law; (3) NIPT, including guidelines, public health insurance, and cost, and (4) managing organizations/consortiums.

## 2. Methods

The country cases were selected based on the following criteria: (1) The country must implement prenatal screening and NIPT test; (2) the guideline and legal configuration information must be easily accessible in English, and (3) the country has introduced universal health insurance, such as Japan.

To identify the relevant studies in Japanese, we searched PubMed, SCOPUS, and Google Scholar databases and the website of CiNii Articles with the following search terms: “NIPT”, “NIPD”, “Non-Invasive Prenatal Test”, and/or “cell-free DNA test”. To identify country cases for guidelines and legal configuration, we used the Google search engine at www.google.com.

The UK, Germany, Italy, Sweden, and Taiwan were selected as comparable countries. All five countries had NIPT guidelines and could be accessed in English. The UK, Germany, and a part of municipalities in Italy are insured for NIPT; in Sweden, the same as in Japan, NIPT guidelines were issued by the academic society [11,12,13]. We could not find any Asian countries that provide NIPT guidelines in English. However, Taiwan was selected because it has promoted the implementation of prenatal screening as a health policy for decades. The United States was excluded from the targeted country due to various abortion laws and perspectives on prenatal diagnosis in each state. An extracted form was used to obtain the following data from policies/guidelines of each selected country: (1) policy on prenatal screening; (2) abortion law; (3) NIPT, including guidelines, public health insurance, and cost, and (4) managing organizations/consortiums.

Ethical approval was not sought for this study because it did not involve human or animal subjects. All included information is freely available in the public domain.

## 3. Results

Country cases were found to have common and different perspectives for all survey topics; the summarized characteristics of included countries are described in Table 1.

### 3.1. Policy on Prenatal Screening

*(a)* 
*Provided information on prenatal diagnosis for all pregnant women*


The UK, Germany, Sweden, and Taiwan had national policies to mandate the HCPs to explain general prenatal screening and obtain informed consent from all pregnant women. Italy did not have a national health policy, but information on prenatal diagnosis has been provided as a customary practice [11,13,20,24,27].

Japanese pregnant women were not officially provided with information on general prenatal diagnosis, including NIPT, related to the notice of the Ministry of Health, Labor, and Welfare (MOHLW) in 1999. Most pregnant women could only receive information on prenatal diagnosis, including NIPT, by obstetrician’s diagnosis or pregnancy handbook issued from the maternity hospital [15].

*(b)* 
*Provided primary prenatal screening for all pregnant women*


The UK, Germany, Italy, Sweden, and Taiwan have policies to offer general prenatal screening during the first trimester. In Japan, MOHLW recommended that all pregnant women perform an ultrasound at least twice in the first trimester. Still, the general prenatal screening for fetal well-being was not mentioned [11,13,15,20,24,27].

### 3.2. Abortion Law: Allowed Abortion for Fetal Chromosomal Abnormality by Law

All six countries possibly allowed abortion within the legal gestation weeks.

Only the UK, Italy, Sweden, and Taiwan allow abortion for fetal abnormalities. The UK enacted the “fetus clause” which allows abortion for fetal abnormalities without any limitation of pregnancy week if the child would suffer from physical or mental abnormalities that would cause them to be severely disabled [16]. In Italy, abortion is allowed in some cases if the child would likely be born with defects or deformities, or if the processes pose a significant risk to the woman’s physical or mental health, such as those linked to severe fetal abnormalities or malformation [22]. Sweden allows abortion if the gestational age is less than eight weeks. If the gestational age is more than eight weeks, Sweden women need permission from the National Board of Health and Welfare to perform an abortion for cases in which the fetus or mother is unhealthy. Sweden also does not allow abortion after 24 weeks of gestational age [25]. Taiwan permits abortion if the risk of teratogenesis may present for the fetus and gestational age is not more than 24 weeks [28].

Germany allows abortion if the gestational age is less than 12 weeks, and pregnant women must prove to their doctor the attendance of the pregnancy-conflict counselling at least three days before the operation. Pregnancy-conflict counselling is averted for pregnancy with danger to life or the threat of severe impairment of pregnant women’s physical or psychological health [19].

Japan allows abortion if pregnancy or delivery may significantly negatively impact the person’s physical health due to bodily or financial reasons. The borderline week of abortion should be less than twenty-two gestational weeks, and the fetus cannot survive its life outside of the mother (MOHLW 1996). Japan does not have a policy regarding abortion due to fetal abnormalities.

### 3.3. NIPT

*(a)* 
*Issued guidelines on NIPT*


Except for Taiwan, Japan and the remaining countries issued guidelines on NIPT. The government issued these guidelines in the UK, Germany, and Italy, and academic societies did in Japan and Sweden.

All guidelines mentioned the common perspectives of NIPT: (1) NIPT is a susceptible and specific test for screening some aneuploidies, particularly for fetal trisomy 21, (2) NIPT is not a diagnostic test and should be followed up by confirmatory tests such as amniocentesis and CVS, and (3) the UK, Germany, Italy and Sweden required taking the combined test (ultrasound and biochemical marker) before NIPT for pregnant women who have certain risks of trisomy 13, 18, and 21 [11,12,13,20]. There was no age restriction for receiving NIPT in those countries. Japan does not require taking the combined test before NIPT, but for pregnant women, the prerequisite is the age of 35 years old. The non-certified facilities in Japan have no age limitation [10].

The processes of NIPT in Japan and the remaining countries are depicted in Figure 1.

*(b)* 
*NIPT fee was covered by public insurance*


In the UK, Germany, Toscana-Italy and Sweden, the NIPT fee will be covered by public health insurance if the combined test reveals that the pregnant woman has a high risk of trisomy 13, 18, or 21 [17,21,30,31]. The NIPT fee is not covered by public insurance in Japan.

*(c)* 
*Most expensive NIPT expenditure*


The expenditure on NIPT in Japan is the most expensive among the other five countries. Prices for NIPT range from $350 to $2900, with the average reported price being around $874 [9].

### 3.4. Managing Organizations/Consortiums

Only Japan formed the NIPT consortium. Japan is the only country that created an accredited system to provide NIPT for the facilities. The committee of the NIPT consortium formed the accredited system to provide NIPT for the facilities. There were ninety-two certified facilities up to 2019 but no certified facilities in eleven out of forty-seven prefectures [9].

## 4. Discussion

Our study compared the policy of prenatal screening, abortion law, and guidelines related to NIPT in some countries: Japan and the UK, Germany, Italy, Sweden, and Taiwan. In addition, the NIPT guidelines are somehow dated to the prenatal diagnosis policy; therefore, the NIPT could be a chance to indulge new steps for prenatal diagnosis in the local market of Japan and the international market.

### 4.1. Policy on Prenatal Screening

Regarding countries, Japan is the only country with no clear policies for prenatal screening issued by the medical authorities. The rest of the countries illustrated the prenatal diagnosis to pregnant women and its policy; then, the women could choose whether to undergo it or not. The lack of national policy led to a disparity in clinical practice in each facility. Hence, the patient may not perceive the comprehensive picture of the NIPT to decide. The lack of national policies also allows the non-certificated facility to provide NIPT or non-specialized HCPs to introduce NIPT results to pregnant women. In summary, malpractice was reported many times, including inappropriate genetic translation of results and lack of post-analysis care that leads to amniocentesis and chorionic villus sampling (CVS) [9]. Japan’s notification on prenatal diagnosis has not been revised since 1999; therefore, the government should revise notification or opinion statements on prenatal diagnosis.

### 4.2. Abortion Law

Japan does not officially allow termination of pregnancy for fetal abnormality. It should be discussed continuously because abortion law is linked to prenatal diagnosis results. Allowing the termination of the abnormal fetus is always controversial and raises many concerns in both academic and non-academic societies. In the UK, pregnant women have the right to decide whether or not to continue their pregnancy [16]. However, the rate of approving abortion for abnormality did not spike, especially in young moms, according to the report by Lee [32]. The law of abortion for fetal abnormality sometimes focuses on the physicians because they need to know that ‘serious abnormality’ is sometimes not mentioned clearly in law. It would be helpful if a list was created to identify which fetal anomalies should be considered legal grounds for abortion.

### 4.3. NIPT

*(a)* 
*Guideline*


The results of our study showed that the NIPT guideline of JSOG was different from the country cases; the main points were the following: (1) Japanese NIPT guidelines did not require any testing before NIPT, and (2) have age restriction for taking NIPT.

Applying NIPT and first-trimester screen tests (FTS) for prenatal screening varies per country. Five included countries imposed the FTS first; then, the NIPT was implemented if the result of FTS was “high risk”, followed by an invasive confirmation test if NIPT is positive. The reason behind this is illustrated by the ratio of advantages and disadvantages of both FTS and NIPT tests. The FTS test is not expensive, and the sensor accuracy is high (90%); nevertheless, it also has a high false-positive value, about 5% [33], which is worrisome. In addition, FTS also requires a good ultrasound machine and a well-trained physician to measure the nuchal translucency. On the other hand, the sensitivity of NIPT is very high; for Down syndrome (trisomy 21) and Edward syndrome (trisomy 18), it is greater than 98%, and for Patau syndrome (trisomy 13), it is greater than 90%. The false-positive rate is low, with 0.1–0.5% in the high-risk population [9]. The evidence states that PPVs for trisomy 21 were more than 93% and 98%, for trisomy 18 were more than 77% and 92%, and for trisomy 13 were more than 43% and 73%, each for 35- and 40-year-old pregnant mother, respectively [34]. However about 1–5% of NIPT results are not reported for various reasons, including the low cell-free fetal DNA in maternal blood. In addition, the NIPT is more expensive than FTS and requires more time to prepare results. The combined strategy minimizes the disadvantages of both tests to provide more accurate and cheaper results.

Because of the value of predicting a fetal aneuploidy in high-risk pregnancy and the low predictive positive value of determining trisomy 21, it was reasonable for JSOG to restrict NIPT to only elderly pregnant women over 35 years old. However, we must also consider the patient rights and wishes besides scientific reasons. All five countries allow pregnant women to choose prenatal screening methods. In a study published in 2019, 235 women who received NIPT at Victorian Clinical Genetics Services were asked about their experience and if they would receive NIPT in the subsequent pregnancy. Ninety-five per cent of participants answered that they had a positive experience and would select NIPT for the subsequent pregnancy. All participants paid the NITP fee by themselves. The first reason for choosing NIPT was to detect a chromosomal abnormality, which 85% of participants selected. The second reason is seeking reassurance, and “peace of mind” was one of the most important reasons for undergoing NIPT, which 56% of participants selected [35]. A recent report also pointed out the use of NIPT for pregnant women under 35 years old. By examining the demographics of all specimens sent to Ariosa Diagnostics, Inc Chen from the US and over other 65 countries, Chen et al. reported that the percentage of specimens submitted by patients under the age of 35 grew from 47.3% in 2014 to 60.3% in 2017 [36].

*(b)* 
*Public insurance and expenditure*


The NIPT fee is not covered by public insurance in Japan; this situation is similar in many other countries. The main reason is that it is costly when compared with FTS. High prices and not being insured for it could prevent pregnant women from selecting the safest and most sensitive method for prenatal screening. In a study published in January 2021, the authors reported that the covered insurance group is more than 3.4 times more likely to take NIPT as an initial screening method than the non-covered insurance group. The study also reported that counsellors would be more likely to recommend NIPT for covered insurance groups [37].

Japan could apply the policy which the UK, Germany, Italy, and Sweden are using: the NIPT fee will be covered by public health insurance if the pregnant woman has a high risk of trisomy 13, 18 and 21 based on the result of the combined test. It would avoid missing high-risk women who cannot receive the NIPT test due to economic reasons. It also helps the government save money because it only has to pay for high-risk pregnant women.

*(c)* 
*Managing organizations/consortium*


Certified facilities in Japan must raise their quality and performance to close the door in front of the non-certified medical facilities. Counselling should provide detailed information on children born with chromosome abnormality; possibility regarding the child’s development in the future and the child’s need for social support. In addition, awareness of prenatal diagnosis and appropriate medical facilities for NIPT is necessary.

### 4.4. Limitation and Imlementation

Our study has limitations. First, comparison is limited to five countries, although there is possibility that another country implements NIPT with universal health insurance available, but it could be included due to the guideline and legal configuration not being easily accessible. Second, most comparisons for the current study comes from European countries which may have different challenges of NIPT implementation from Asian countries due to cultural, demographic, and socioeconomic background that may affect policy, law, and the practice of prenatal screening. Therefore, it is difficult to draw a conclusion regarding the implementation if the comparison is too different.

Further study needs to explore policy on prenatal screening information and insurance coverage of NIPT for pregnant mothers, especially by comparing data with Asian countries. Japan does not currently perform prenatal screening information delivery as customary practice. Pregnant mothers potentially do not know about it and its importance; they may feel dissatisfied and disappointed if they realize they obtain the information later, only after the abnormalities are suspected by the doctor, and there may be potential late decision to receive NIPT, or if abnormalities are detected after NIPT but the pregnancy is already progressed beyond 22 weeks. Therefore, information about prenatal screening is important to be delivered early for a pregnant mother. NIPT fee is too high in Japan compared to countries in the current study; more comparison is needed, especially regarding information on whether NIPT can be covered by insurance in order to serve as reference for consideration to improve Japan’s policy regarding prenatal screening information and fee coverage by health insurance.

## 5. Conclusions

Pregnant women in Japan suffer from numerous drawbacks, including the lack of information, the high cost, the long distance between them and the certified facilities, the availability of non-certified centres, the lack of general scanning for fetal abnormality at first trimester, the disparity in the practice of NIPT guidelines between the certified and the non-certified facilities, and the NIPT fee that is not covered by public insurance. Pregnant women-centered prenatal diagnosis policy, including NIPT, should be established in Japan by learning cases from other countries. The Japanese government should appoint the leadership to develop the NIPT policy in collaboration with many stakeholders other than JSOG. The medical professionals should support pregnant women to self-determine whether to receive prenatal diagnosis and NIPT after providing comprehensive information to all pregnant women.

## Figures and Tables

**Figure 1 ijerph-19-16404-f001:**
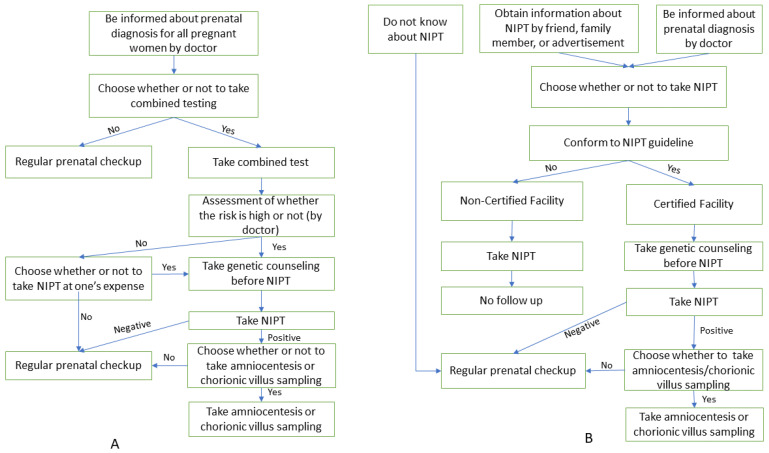
The process of NIPT in Japan and other countries. (**A**) Process of NIPT in other countries; (**B**) Process of NIPT in Japan.

**Table 1 ijerph-19-16404-t001:** Summarized characteristics of country cases.

		Japan [9,10,14,15]	UK [11,16,17]	Germany [13,18,19]	Italy [20,21,22]	Sweden [12,23,24,25,26]	Taiwan [27,28,29]
(1) Policy on prenatal screening	Provided information on prenatal diagnosis for all pregnant women	No	Yes	Yes	Yes	Yes	Yes
	Provided primary prenatal screening for all pregnant women	No	Yes	Yes	Yes	Yes	Yes
(2) Abortion law	Allowed abortion for fetal chromosomal abnormality by law	No	Yes	Yes	No	Yes	Yes
(3) NIPT	(1)-a. Guideline (government)	No	Yes	Yes	Yes	No (evaluated report)	No
	(1)-b. Guideline (academic society)	Yes	No	No	No	Yes	No
	(2) Public health insurance for NIPT	No	Yes	Yes	Yes (partially)	Yes	No
	(3) Cost for NIPT (trisomy 13, 18 and 21) without insurance	Certified facilities 190,000 yen	40,000–60,000 yen	No information	30,000–50,000 yen	50,000–60,000 yen	80,000–130,000 yen
		Non-certified facilities 160,000 yen					
(4) Managing organization/consortium	Formed NIPT consortium	Yes	No	No	No	No	No

## Data Availability

Not applicable.

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
