# Peer review of "Non-Invasive Prenatal Testing (NIPT) Implementation in Japan: A Comparison with the United Kingdom, Germany, Italy, Sweden, and Taiwan"

_ijerph, 2022, doi:10.3390/ijerph192416404_

Round 1

Reviewer 1 Report (New Reviewer)

Your study is about NIPT implementation in several countries. A valuable scientific contribution of your study would have been the evaluation of the differences regarding rate detection of aneuploidies not only implementation of NIPT.

Please clearly mention if Japan offers general prenatal screening with first trimester anomaly scan at 11-13 weeks. If not, than this should be the first conclusion of your study.

If NIPT was not covered by public insurance than this is the cause for non-certified centers. Another conclusion should be that State should support NIPT for high risk genetic pregnancies detected by usual first trimester screening.

Conclusion section should be revised and contain a take home message for clinical practitioners

Author Response

Comment 1: Your study is about NIPT implementation in several countries. A valuable scientific contribution of your study would have been the evaluation of the differences regarding rate detection of aneuploidies not only implementation of NIPT.

Reply: We thank the reviewer for valuable comment to improve our manuscript. We do agree with the point of view of the reviewer. We did systemic research on the rate detection of aneuploidies in the UK, Germany, Italy, Taiwan, and Sweden. However, we could not find relevant data yet except one single-center study data in the UK and one in Germany. Therefore, we decided that we only focus on the implementation of the NIPT.

Comment 2: Please clearly mention if Japan offers general prenatal screening with first trimester anomaly scan at 11-13 weeks. If not, than this should be the first conclusion of your study.

Reply: Japan does not offer general prenatal screening at first trimester for anomality. We have mentioned it in session 3.1 Policy on prenatal screening.  We also added this information on the conclusion as reviewer suggested:

We edit these sentences: “Pregnant women in Japan suffer from numerous drawbacks, including the lack of information, the high cost, the long-distance between them and the certified facilities, the availability of non-certified centers, and the disparity in the practice of NIPT guidelines between the certified and the non-certified facilities”.

Change to: “Pregnant women in Japan suffer from numerous drawbacks, including the lack of information, the high cost, the long-distance between them and the certified facilities, the availability of non-certified centers, the lack of general scanning for fetal abnormality at the first trimester, the disparity in the practice of NIPT guidelines between the certified and the non-certified facilities, and the NIPT fee was not covered by public insurance” (line 296 – 300, page 8).

Comment 3: If NIPT was not covered by public insurance than this is the cause for non-certified centers. Another conclusion should be that State should support NIPT for high-risk genetic pregnancies detected by usual first trimester screening.

Reply: We thank the reviewer for pointing this out, the suggestion is valuable, and we provided it in discussion section as: “Japan could apply the policy which the UK, Germany, Italy, and Sweden are using: the NIPT fee will be covered by public health insurance if the pregnant woman has a high risk of trisomy 13, 18 and 21 based on the result of the combined test. It would avoid missing high-risk women who cannot take the NIPT test due to economic reasons. It also helps the government save money because it only has to pay for high-risk pregnant women.” (Line 261 – 266, page 7).

We also edited the conclusion, added the statement that NIPT was not covered by public insurance in Japan as the reviewer suggested:

We edit these sentences: “Pregnant women in Japan suffer from numerous drawbacks, including the lack of information, the high cost, the long-distance between them and the certified facilities, the availability of non-certified centers, and the disparity in the practice of NIPT guidelines between the certified and the non-certified facilities”.

Change to:Pregnant women in Japan suffer from numerous drawbacks, including the lack of information, the high cost, the long-distance between them and the certified facilities, the availability of non-certified centers, the lack of general scanning for fetal abnormality at first trimester, the disparity in the practice of NIPT guidelines between the certified and the non-certified facilities, and the NIPT fee was not covered by public insurance”. ((line 296 – 300, page 8).

Comment 4: Conclusion section should be revised and contain a take home message for clinical practitioners.

Reply: We are so grateful for your guidance; we have added a take home message for clinical practitioners: “The medical professionals should support pregnant women to self-determine whether take prenatal diagnosis and NIPT after providing comprehensive information to all pregnant women” (line 304 – 306, page 8).

Reviewer 2 Report (New Reviewer)

Thank you for the opportunity to review the manuscript entitled “Non-Invasive Prenatal Testing (NIPT) implementation in Japan: A comparison with the United Kingdom, Germany, Italy, Sweden, and Taiwan”. The authors aimed to identify the Japanese challenges of adopting NIPT into prenatal diagnosis by comparing the system and process with other  countries: The United Kingdom, Germany, Italy, Sweden, and Taiwan. The study is interesting and should be of interest of the Journal readers. However, I have some remarks:

-       Abstract: “The Non-Invasive Prenatal Testing (NIPT) guideline was issued and applied in 2013 by the Japanese Medical Association.” – the sentence is not informative, was NIPT introduced as funded or commercial? What in fact changed in 2013? Was it just approved and available to all in Japan? Was NIPT available earlier as only commercial?

-       Introduction: 

“For instance, the NIPT is performed only after a positive result of traditional prenatal scanning.” – what kind of traditional scanning? Do you mean “is performed for free/without costs for a patient”? In my country for instance, it is never paid by NHS and even if a patient performs it on her own, a diagnosis must be confirmed by traditional methods /amniocentesis/ - are there any other countries with such approach? Such point of view should also be presented.

“The non-certified facilities (NCFs) were allowed to perform NIPT 2016. “ – “in/from 2016”. 

-       Results –  “Provided primary prenatal screening for all pregnant women” – you have not mention differences in selected countries recommendations regarding mothers’ age, I am not sure if there are any..but e.g. in my country routine screening in women over 35 years includes amniopuncture 

-       It would be interesting and very useful to add a table including a summary of academic guidelines for NIPT in the analyzed countries (for whom? When? Etc.)

-       “Japan did not require taking the combined test before NIPT but required only pregnant women over thirteen or five years old could take NIPT officially at the certified facilities.” – I do not understand this sentence – what these ages relate to?

-       Discussion: “Japan was not officially allowed fetal abnormality” – probably there was a problem with English translation here…but this sounds “ambiguously”… Similarily, there are a lot of “problematic” sentences, e.g. “Detailed information regarding the  baby's development and the child's social support with chromosome abnormality during counselling.”…

-       Also, there is a number of typos, e.g. “Limitation and imlementation”

Needs proof-reading /English editing/!

Author Response

Comment 1:  Abstract: “The Non-Invasive Prenatal Testing (NIPT) guideline was issued and applied in 2013 by the Japanese Medical Association.” – the sentence is not informative, was NIPT introduced as funded or commercial? What in fact changed in 2013? Was it just approved and available to all in Japan? Was NIPT available earlier as only commercial?. 

Reply: We are so grateful for your guidance. Our intention is to emphasize when the NIPT guideline was issued and applied. The further explanation has been provided in the second paragraph of the introduction session (line 46 – 52, page 1). We have edited the first sentence of abstract session in order to make it clear and more informative:

We edit these sentences: “The Non-Invasive Prenatal Testing (NIPT) guideline was issued and applied in 2013 by the Japanese Medical Association”.

Change to: “The Non-Invasive Prenatal Testing (NIPT) guideline was issued and applied in 2013 by the Japanese Medical Association.  Since issued, NIPT practice in Japan still have some problem related to indication, access, cost coverage and uniformity. Therefore, our study aimed to identify the Japanese challenges of adopting NIPT into prenatal diagnosis by comparing the system and process with other countries”. (Line 15 – 18, page 1).

Comment 2: Introduction: “For instance, the NIPT is performed only after a positive result of traditional prenatal scanning.” – what kind of traditional scanning? Do you mean “is performed for free/without costs for a patient”? In my country for instance, it is never paid by NHS and even if a patient performs it on her own, a diagnosis must be confirmed by traditional methods /amniocentesis/ - are there any other countries with such approach? Such point of view should also be presented.

Reply: Thank you for pointing this out. According to the article we cited (reference no. 6), “traditional prenatal scanning” means enhanced first-trimester screening (eFTS) or maternal serum screening. We agree that using “traditional prenatal scanning” makes the reader confused. Therefore, we have edited the sentence to make it clearer.

We edit these sentences: “For instance, the NIPT is performed only after a positive result of traditional prenatal scanning. Using NIPT as a second-tier is cost-saving compared to using conventional prenatal scanning methods, but more cases of chromosomal anomalies would be detected when NIPT was implemented as a first-tier test [6]”.

Change to: “For instance, the NIPT is performed only after a positive result of enhanced first-trimester screening or maternal serum screening. Using NIPT as a second-tier is cost-saving compared to using conventional prenatal scanning methods, but more cases of chromosomal anomalies would be detected when NIPT was implemented as a first-tier test [3,6]”. (Line 41-45, page 1).

Comment 3: “Provided primary prenatal screening for all pregnant women” – you have not mention differences in selected countries recommendations regarding mothers’ age, I am not sure if there are any... but e.g., in my country routine screening in women over 35 years includes amnio puncture. 

Reply: Thank you for your comment. We think the reviewer may be misinterpreting our message. In this session (policy on prenatal screening), we just only focused if the selected countries have any policy to provide primary prenatal screening for pregnant women or not. That is why we did not report the differences in selected countries’ recommendations regarding mothers’ age.

Comment 3:   It would be interesting and very useful to add a table including a summary of academic guidelines for NIPT in the analyzed countries (for whom? When? Etc.).

Reply: Thank you for pointing this out, we agree that the mentioned information is very useful and important, and we also intended to add it on the summary table. However, we found out that the process of NIPT in the analyzed countries is very similar, and we think that using table to summarize it will be lengthy and difficult to follow. Therefore, we decided to depict the process of NIPT in figure 1 (Page 5), which included the information that you mentioned.

Comment 4: “Japan did not require taking the combined test before NIPT but required only pregnant women over thirteen or five years old could take NIPT officially at the certified facilities.” – I do not understand this sentence – what these ages relate to?

Reply: We are so thankful for your comment, this is a typo error, therefore, we have corrected the sentence.

We edit this sentence: “Japan did not require taking the combined test before NIPT but required only pregnant women over thirteen or five years old could take NIPT officially at the certified facilities”.

Change to: “Japan do not require taking the combined test before NIPT, but for pregnant women, the prerequisite is the age that should be over 35 years old.” (Line 155-158, page 4).

Comment 5: Discussion: “Japan was not officially allowed fetal abnormality” – probably there was a problem with English translation here…but this sounds “ambiguously”… Similarly, there are a lot of “problematic” sentences, e.g., “Detailed information regarding the baby’s development and the child's social support with chromosome abnormality during counselling.

Reply: Thank you for pointing this out. We agree with this comment. Therefore, we have corrected these sentences.

We edit this sentence: “Japan was not officially allowed fetal abnormality”.

Change to: “Japan was not officially allowed termination of pregnancy for fetal abnormality” (line 200, page 6).

We edited this sentence: “Detailed information regarding the baby's development and the child's social support with chromosome abnormality during counseling”.

Change to: “Counselling should provide detailed information of children born with chromosome abnormality; possibility regarding the child's development in the future and the child's need of social support” (line 269-270, page 7).

Round 2

Reviewer 1 Report (New Reviewer)

all suggestions have been implemented

This manuscript is a resubmission of an earlier submission. The following is a list of the peer review reports and author responses from that submission.

Round 1

Reviewer 1 Report

First of all, I would like to thank you the opportunity to revise the manuscript entitled "Non-Invasive Prenatal Testing (NIPT) implementation in Ja-2 pan: A comparison with the United Kingdom, Germany, Italy, 3 Sweden, and Taiwan". Despite the originality of the topic, I cannot support the publication of this work in IJERPH. The interest of the topic is restricted mainly to Japan, so I would suggest its publication in a more local/regional scientific journal. In addition, the manuscript should be improved in various aspects. 

First of all, the English language should be extensively edited. Probably, some my concerns about it are a consequence of my difficulties to understand the text. I am not a native english speaker, but I think that the manuscript would improve if the English language was reviewed. 

Second, there are basic issues that would need to be fixed.

  • Abstract: the objective does not match with the conclusions. 
  • The introduction: this part should somehow explain the reason for conducting the study. In my opinion, the introduction of this manuscript does not highlight the problems or issues of NIPT in Japan, not at least in a comprehensive manner.
  • Materials and methods: the heading should be change to methods. The country inclusion criteria are not well explained (I would highly reccomend to add a flow chart). Besides, the authors do not explain or why they select the points that are studied ( (1) policy on prenatal screening; (2) abortion law; (3) NIPT, including guideline, public health insurance, and cost; and (4) managing organizations/consortiums).  
  • Results: the headings of the points that have been selected as endpoints of the study do not match with the ones from the table 1. In this table, ir would be very helpful to have other currencies such as euros or US dollars in order to understand the cost of the NIPT in each country. Figure 1 is very helpful to understand the prenatal screening process in Japan, since the text is difficult to understand, and I do not think that is even explained in detail before the discussion. 
  • Discussion: I would not divide it in different sections, since they are generally short, and it would be easier to follow.  In addition, the first paragraph says that "Our investigation hailed numerous differences in the perception....", but the study is not about perception, it is an objective study that tries to identify the diferences of NIPT between Japan and other countries.

I think that the manuscript could be published after performing major changes in a more local scientific journal.

Reviewer 2 Report

Authors performed an interesting study to evaluate the implementation of NIPT In different countries. Indeed, almost 10 years ago NIPT started to be used out of research settings but the uptake of this tests has had different velocity among different countries, also according to different regulatory rules and scientific/medical guidelines.

The idea of this study is nice and the result could be useful.

However, the manuscript suffers from many flaws.

The introduction should be better organized, especially the first part introducing the NIPT.

The first sentence should be changed since you cannot start saying “it” without saying what is….

In addition, from line 33 to line 43 the NIPT could be better presented and described. In addition, I did not understand when author said that NIPT could detect trisomies 13,18,21 and 22….. usually it is described as able to detect the major trisomies (13,18,21) but also any other chromosomal imbalance of autosomal chromosomes and sexual chromosomes. 

Results:

There is no point to add in any subsection the references in the subtitle. Please delete the references from there and put it only throughout the text.

Figure 1 is not easy to read because you need to zoom a lot. See if characters could be put a little bigger.

I do not agree about 45% of positive predictive value for trisomy 21. See

Sasaki Y, Yamada T, Tanaka S, Sekizawa A, Hirose T, Suzumori N, Kaji T, Kawaguchi S, Hasuo Y, Nishizawa H, Matsubara K, Hamanoue H, Fukushima A, Endo M, Yamaguchi M, Kamei Y, Sawai H, Miura K, Ogawa M, Tairaku S, Nakamura H, Sanui A, Mizuuchi M, Okamoto Y, Kitagawa M, Kawano Y, Masuyama H, Murotsuki J, Osada H, Kurashina R, Samura O, Ichikawa M, Sasaki R, Maeda K, Kasai Y, Yamazaki T, Neki R, Hamajima N, Katagiri Y, Izumi S, Nakayama S, Miharu N, Yokohama Y, Hirose M, Kawakami K, Ichizuka K, Sase M, Sugimoto K, Nagamatsu T, Shiga T, Tashima L, Taketani T, Matsumoto M, Hamada H, Watanabe T, Okazaki T, Iwamoto S, Katsura D, Ikenoue N, Kakinuma T, Hamada H, Egawa M, Kasamatsu A, Ida A, Kuno N, Kuji N, Ito M, Morisaki H, Tanigaki S, Hayakawa H, Miki A, Sasaki S, Saito M, Yamada N, Sasagawa T, Tanaka T, Hirahara F, Kosugi S, Sago H; Japan N. I. P. T. Consortium. Evaluation of the clinical performance of noninvasive prenatal testing at a Japanese laboratory. J Obstet Gynaecol Res. 2021 Oct;47(10):3437-3446. doi: 10.1111/jog.14954. Epub 2021 Aug 5. PMID: 34355471.

In addition, in line 205 authors refer to “Lee[7]” but reference n. 7 is not from any Lee. Please correct.

Discussion should be largely revisited: apart from commenting on the differences and similarities among the nations which are objects of the study, it needs a strength and limitations section, and an implementation/future development section. You could benefit of these papers to implement your manuscript:

- Salvesen KÅB, Glad R, Sitras V. Controversies in implementing non-invasive prenatal testing in a public antenatal care program. Acta Obstet Gynecol Scand. 2022 Mar 24. doi: 10.1111/aogs.14351. Epub ahead of print. PMID: 35332520.

- Gadsbøll K, Petersen OB, Gatinois V, et al. Current use of noninvasive prenatal testing in Europe, Australia and the USA: a graphical presentation. Acta Obstet Gynecol Scand. 2020;99:722-730.

- Suzumori N. What are the ethical issues involved in noninvasive prenatal testing in Japan? J Obstet Gynaecol Res. 2022 Feb;48(2):300-305. doi: 10.1111/jog.15089. Epub 2021 Nov 2. PMID: 34729844.

- Perrot A, Horn R. The ethical landscape(s) of non-invasive prenatal testing in England, France and Germany: findings from a comparative literature review. Eur J Hum Genet. 2021 Oct 4. doi: 10.1038/s41431-021-00970-2. Epub ahead of print. PMID: 34602609.

- Yamamoto K, Chang H, Fukushima A. Pregnant women's experiences of non-invasive prenatal testing (NIPT) in Japan: A qualitative study. J Genet Couns. 2022 Apr;31(2):338-355. doi: 10.1002/jgc4.1494. Epub 2021 Aug 25. PMID: 34432354.

- take into consideration also your reference n.1 from Allyse et al.

Last but not least, the manuscript suffers from poor English language quality. Please ensure to have it adequately improved by English fluent speaker/writer.